# The use and impact of virtual reality programs supported by aromatherapy for older adults: A scoping review protocol

Lillian Hung[1,2]* , Joey Wong[2], Haopu Ren[2], W. Ben Mortenson[3], Angelica Lim[4], Jennifer Boger[5,6], Christine Wallsworth[2], Yong Zhao[2]

**1** School of Nursing, University of British Columbia, Vancouver, British Columbia, Canada, **2** IDEA Lab, University of British Columbia, Vancouver, BC, Canada, **3** International Collaboration on Repair Discoveries, GF Strong Rehabilitation Research Program, Department of Occupational Science and Occupational Therapy, Faculty of Medicine, University of British Columbia, Vancouver, BC, Canada, **4** School of Computing Science, Simon Fraser University, Vancouver, BC, Canada, **5** Department of Systems Design Engineering, University of Waterloo, Waterloo, ON, Canada, **6** School of Health and Exercise Sciences, University of British Columbia (Okanagan), Kelowna, BC, Canada

☯ These authors contributed equally to this work.
* lillian.hung@ubc.ca

**Data Availability Statement:** No datasets were generated or analysed during the current study. All relevant data from this study will be made available upon study completion.

## Abstract

Both virtual reality and aromatherapy have shown promising impacts on the health and well-being of older adults. Aromatherapy has been reported to enhance immersive experiences during virtual reality programs. However, studies on the combined use and impact of virtual reality and aromatherapy for older adults have not been systematically reviewed. Therefore, this scoping review will identify existing types of virtual reality programs supported by various forms of aromatherapy and their outcome measures and results on the well-being of older adults. This review will be conducted in accordance with the Joanna Briggs Institute methodology or scoping reviews and will be reported in line with the Preferred Reporting Items for Systematic Reviews and Meta-Analyses extension for Scoping Reviews. The search strategy will encompass both published and unpublished papers. The databases to be searched are CINAHL, MEDLINE, Embase, Scopus, Web of Science, ACM digital library, IEEE Xplore digital library, Compendex, ProQuest, and Google Scholar. Two independent reviewers will perform title and abstract screening, full-text screening, and data extraction. Data analysis and synthesis will be discussed by the whole research team, mapped in the literature table and accompanied by a narrative summary. Scoping review data will be collected from publicly available articles; research ethics approval is not required. The findings will be disseminated through a peer-reviewed publication and conference presentations.

## Introduction

### Virtual reality applications for older adults

Virtual reality (VR) technology has emerged as a transformative tool with the potential to enhance the health and well-being of older adults, particularly those residing in long-term care

**Funding:** This scoping review is supported by funding from Brain Canada in collaboration with Alzheimer Society of Canada (Grant number: GR028233; https://alzheimer.ca/en/research/alzheimer-society-research-program/latest-funding-results) awarded to LH. The sponsor does not play any role in the study design, data collection and analysis, decision to publish, or preparation of the manuscript.

**Competing interests:** The authors have declared that no competing interests exist.

settings, which often grapple with the challenge of limited human resources [1]. VR has been defined in MeSH headings as "Using computer technology to create and maintain an environment and project a user's physical presence in that environment, allowing the user to interact with it." [2] VR offers a dynamic platform that transcends traditional boundaries, providing immersive experiences that can address a myriad of physical, cognitive, psychological, and social well-being needs [3]. Through motion-based exergames, immersive environments, and interactive simulations, VR interventions hold promise in improving physical health outcomes such as balance, mobility, and attention among older adults [4–6]. Furthermore, VR presents opportunities for cognitive stimulation through tailored exercises and activities designed to enhance memory, problem-solving skills, and overall cognitive function [7, 8]. Moreover, VR experiences can facilitate social engagement and interaction, mitigating feelings of loneliness and isolation commonly experienced by older adults in care settings [9]. As VR technology continues to evolve and become more accessible, its potential to revolutionize how we approach aging and care for older adults becomes increasingly apparent, offering new avenues for promoting healthy aging and improving quality of life in later years. For example, the individualized and immersive nature of VR interventions holds significant promise in providing older adults with engaging and error-free self-training opportunities, empowering them to participate in their own health and well-being actively [1]. Among various technologies aiming to enhance the immersive effect of VR, multisensory VR, especially integrating olfactory stimuli, has shown outstanding advantages. This use of olfactory stimuli aligns closely with the principles of aromatherapy, a therapeutic that leverages scents to influence emotional and cognitive states, making it a natural complement to VR applications.

## Aromatherapy and older adults

Aromatherapy is a complementary therapy using fragrance or smell [10] and often with "a controlled and therapeutic use of essential oils" [11]. Essential oils are extracted from the "leaves, flowers, stems, fruit, seeds, bark and roots of a range of aromatic plants." [12] Aromatherapy using fragrances like cedar, cinnamon, chocolate, lavender, and various essential oils has been used for older adults to improve sleep quality [13], relieve pain [14], reduce their depression and anxiety levels [14, 15], improve cognitive functions and memory recall, potentially easing the burden of nursing care [16]. There are different ways that aromatherapy has been adopted for older adults, for example, through inhalation [17] and massage [15]. Although aromatherapy has been shown to be promising with a potential positive impact on older adults, a review by Ball et al. [18] and a report from the National Institute for Health and Care Excellence [19] highlighted uncertainties in how and to what extent aromatherapy could benefit people living with dementia, such as improving depressive symptoms or having an impact on behaviours. Some studies related to aromatherapy and dementia were reported with inconsistent study designs and outcome measurements [18].

## Integration of aromatherapy in VR programs

Aromatherapy can significantly enhance VR experiences by engaging the olfactory senses, thereby creating a more immersive and multi-sensory environment [20]. Due to the specific connection between the sense of smell and brain areas processing memory, emotions and associative learning, specific scents in VR can evoke emotions, trigger memories, and enhance the realism of virtual settings, making the experience more vivid and impactful [21]. For instance, the scent of pine in a VR forest simulation can heighten the sense of being in a real forest, while calming lavender can augment relaxation in a virtual meditation session [22, 23]. This integration not only deepens the sensory engagement but also has potential therapeutic benefits, reducing stress and enhancing

mood, decreasing systolic blood pressure and heart rate, increasing vigor and decreasing negative emotions like tension, confusion, distress, anger, fatigue, and depression, making VR applications in therapy, gaming, and education more effective and enjoyable [22, 23]. Additionally, incorporating aromatherapy in VR programs can also improve older adults' psychological health, including happiness, sleep quality, and life satisfaction [24]. Thus, the integration of aromatherapy into VR might potentially hold a promising future.

Several systematic reviews have explored the impacts of VR [1, 3, 6, 8] or the impacts of aromatherapy [13, 18, 25] for common older adults [1, 3, 6, 13, 25] or residents living with dementia in isolation [8, 18]. One systematic review highlighted that VR interventions have been effective in improving physical health outcomes in older adults [1]. Another review noted that immersive VR games, particularly motion-based exergames, are among the most popular VR interventions for older adults. Research on these games often focuses on balance, attention, confidence, and interaction, which correspond to physical, cognitive, psychological, and social functions, respectively [3]. The third review demonstrated that immersive VR shows promise as a complementary tool for older adults in health, rehabilitation, and active aging. None of the three systematic reviews mentioned the use of aromatherapy in conjunction with VR, though they provide information on outcome measures and results [6]. In contrast, two systematic reviews demonstrated that aromatherapy can enhance sleep quality and reduce stress, pain, anxiety, depression, and fatigue in older adults, but did not discuss VR [13, 25]. Besides, one systematic review summarized all the outcome measures and results addressed the benefits of aromatherapy for people with dementia: agitation assessed by the Cohen-Mansfield Agitation Inventory (CMAI), mood assessed by the Cornell Scale for Depression in Dementia (CSDD), quality of life assessed by Blau Quality of Life [18]. Although the author cannot draw definitive conclusions due to the low quality of evidence, some studies have reported positive effects of aromatherapy on agitation and behavioural issues and mental health.

There is only one systematic review that has summarized the combination of VR with complementary alternative medicines, which include meditation, hypnosis, Taichi, Qigong, Yoga, and aromatherapy [26]. Specifically, only one study has written about this integration of aromatherapy in a VR program for older adults, demonstrating significant improvements in happiness, sleep quality, perceived stress, meditation experience, and life satisfaction [24].

Given the recent explosive development of innovative technology, the application of VR supported by aromatherapy for older adults has generated considerable interest, thus motivating the need for this scoping review. The novelty of this scoping review lies in addressing this knowledge gap in the current literature. The objectives of the scoping review are to (1) comprehensively understand and summarize the extent of evidence regarding the existence of VR programs supported by aromatherapy for older adults, and (2) evaluate the reported outcome measures and results studied and reported in the literature.

## Review questions

1. How have VR programs for older adults been supported by aromatherapy?

2. What outcome measures and results of VR programs for older adults supported by aromatherapy have been reported in the literature?

## Contributions

- Addressing a knowledge gap: provides an innovative synthesis of evidence regarding the integration of VR and aromatherapy for older adults, addressing a gap in current literature.

- Novel Approach to well-being of older adults: Highlights the potential of combined VR and aromatherapy interventions to enhance well-being in older adults, offering a novel option.

- Guiding future research: Offers foundational insights to inform the design and implementation of future studies focused on this combination of intervention.

- Nursing practice applications: provides practical insights that can influence nursing practices and care strategies for older adults, particularly in improving special aspects of their quality of life.

## Methods

### Design

The proposed scoping review will be carried out following the Joanna Briggs Institute (JBI) methodology for scoping reviews [27]. This methodology was selected for its comprehensive framework, which systematically includes a variety of study designs and emphasizes delivering practical evidence for healthcare practice and policy development. This ensures a thorough and transparent approach to synthesizing the available literature. The scoping review is particularly suitable for our research questions as it allows for a wide-ranging exploration of the emerging field of VR and evaluates the extent of research activity to inform practice and future research.

### Inclusion criteria

**Participants.**   This review will focus on studies examining the combined use of aromatherapy and VR for older adults. Studies that address only other age groups will be excluded. Despite the heterogeneity in physical, psychological, cognitive, and social functions among older adults, which cannot be fully captured by chronological age [28], aging is typically measured by chronological age. Conventionally, individuals aged 65 years or older are referred to as 'older adults' [29–31]. This scoping review will adhere to the widely recognized criteria adopted by the World Health Organization (WHO) and the United Nations (UN) in their epidemiological reports, considering individuals aged 65 and above as older adults.

**Concept.**   Our review will focus on studies utilizing fully immersive VR technology combined with olfactory stimuli for older adults. Olfactory cues have been reported to enhance the immersive experience of VR by increasing the sense of spatial presence and realism, thereby affecting participants' physiological and psychological responses [32] as well as their affective and behavioural reactions [20]. Thus, we will include studies addressing both VR and olfactory stimuli simultaneously and exclude those focusing only on one of these two fields. We will exclude studies using less immersive technology, such as 2D simulations and virtual environment-based VR interventions. Mixed reality (MR), which blends physical and digital worlds where physical and digital objects coexist and interact in real-time, can be achieved with head-mounted displays [33]. Due to its highly immersive and interactive nature, this scoping review will include studies involving mixed reality programs alongside VR. Conversely, augmented reality primarily engages users through a 2D flat screen and offers a less immersive and interactive experience [33]. Therefore, this scoping review will not include studies focused solely on augmented reality.

**Context.**   This review will include literature addressing the application of VR enhanced by aromatherapy for older adults in all settings, including hospitals, long-term care homes, assisted living settings, communities, and private homes, regardless of countries and cultural or ethnical backgrounds.

**Types of studies.** This scoping review will include quantitative, qualitative, and mixed research designs. In addition to experimental and quasi-experimental study designs, we will consider analytical observational studies, prospective and retrospective cohort studies, case-control studies, and cross-sectional studies. Descriptive observational study designs, such as case series, individual case reports, and descriptive cross-sectional studies, will also be included. Qualitative study designs like phenomenology, grounded theory, ethnography, qualitative description, action research, and feminist research will be considered. Systematic reviews and text and opinion papers will also be included.

## Research team for the scoping review

The research team comprises student trainees, interdisciplinary academic scholars from nursing, occupational medicine, and computer science, and two patient partners. This diverse team composition has enriched the researchers' understanding of the topic and underscored the significance of incorporating perspectives from lived experiences. All team members, including the patient partners, will participate in discussions and analyses of data from the selected studies during research meetings.

## Search strategy

The search aims to identify both published and unpublished studies. A preliminary search in the MEDLINE database and Google Scholar was conducted to pinpoint articles related to the topic. Terms found in article titles, abstracts, and index terms were used to formulate a comprehensive search strategy for English-based search engines, including CINAHL, MEDLINE, Embase, Scopus, Web of Science, ACM Digital Library, IEEE Xplore Digital Library, Compendex, ProQuest, and Google Scholar (first 200 results). The search strategy will follow logical expressions adjusted for each database's search tips on auto-stemming, wildcard, truncations, and quotations: Title/Abstract/Keywords/Subject Headings ((older adult OR senior OR elderly) AND (aromatherapy OR essential oil OR fragrance OR odor OR olfactory OR scent OR smell) AND (virtual reality OR mixed reality OR VR)). A detailed search strategy for MEDLINE is included in the S1 Appendix. Additionally, the reference lists of all included sources will be examined for additional relevant studies.

We are collaborating with two medical librarians to refine the search strategy to ensure we capture all relevant and key articles. As a scoping review, we aim to include all potential articles. For example, we will include articles involving participants both younger and older than 65 years and manually extract relevant data during the extraction stage. The inclusion criteria encompass studies in any language with English abstracts, with a cut-off date of May 2024. However, only full-text articles in English and Chinese will undergo full-text screening due to the language proficiency of the team members. Unpublished studies and grey literature will be explored through Google Scholar. Conference abstracts will be used to establish contact with study authors to access full-text studies potentially. There will be no limitation based on publication date.

## Study/source of evidence selection

All identified citations will be collected and uploaded into the Covidence systematic review tool (Veritas Health Innovation Ltd, AU) [34], where duplicates will be removed. After a pilot test to ensure consistency in evidence selection among the review team, two independent reviewers will screen titles and abstracts against the inclusion criteria. Sources deemed potentially relevant will be retrieved in full, and their citation details will be imported into the Covidence tool. Two independent reviewers will then perform a thorough assessment of the full

texts against the inclusion criteria. Reasons for excluding sources at the full-text stage will be documented and included in the scoping review report. Any disagreements between reviewers will be resolved through group discussions. The outcomes will be detailed comprehensively in the final scoping review and illustrated in a flow diagram based on the Preferred Reporting Items for Systematic Reviews and Meta-Analyses extension for scoping reviews [35].

## Data extraction

Data extraction from the selected articles will be conducted by two independent reviewers using the Garrard Matrix method, which involves inputting data into a designated spreadsheet that captures detailed information such as participants, study designs, interventions, outcome measures, results, impacts, and key findings relevant to the review questions [36]. A draft extraction form is available in the S2 Appendix.

Following the JBI methodology, we will conduct a pilot test of the extraction tool on three full-text articles to ensure its reliability, with results checked for consistency. Since the goal of this scoping review is to map the existing literature landscape rather than critically evaluate the evidence, we will not assess the methodological quality of the studies. The draft spreadsheet will be adjusted and revised as necessary based on discussions within the team during the data extraction phase, with any modifications documented in the scoping review reports. If required, authors of studies will be contacted to obtain missing or additional data.

## Data synthesis

Extracted data will be organized into a literature table and accompanied by a narrative summary that ties the findings back to the study objectives and research questions. This narrative summary will organize the tabulated results into thematic categories relevant to the scope of the review. Our comprehensive review will present both qualitative and quantitative data (e.g., happiness, perceived stress, sleep quality, and life satisfaction), ensuring a thorough examination of the existing literature on the topic. To enhance clarity, we will structure the findings into two distinct sections that address the review questions directly. The section titled "Interventions" will review question 1, "How have VR programs for older adults been supported by aromatherapy?" which will focus on the characteristics, design, and implementation of these interventions. The section titled "Outcome measures and results" will correspond to review question 2, "What outcome measures and results of VR programs for older adults supported by aromatherapy have been reported in the literature?" This section will present the findings on the assessment strategies and reported impacts on the well-being of older adults.

## Patient and public involvement

Christine Wallsworth, a patient partner, has been integral to the conception and planning of the scoping review and will continue to be actively involved in the literature screening and data analysis stages. As an older adult, her firsthand experiences provide invaluable insights, particularly concerning VR programs and aromatherapy. Her perspectives will be shared with the entire review team and incorporated into our discussions. Additionally, Christine will participate in staff meetings where the main findings are disseminated. In recognition of her valuable contributions and time, she will be invited to co-author the scoping review report article.

## Ethics and dissemination

For this scoping review, data will be sourced from publicly available articles, eliminating the need for informed consent or research ethics approval. The findings will be shared with

healthcare practitioners and the public through a peer-reviewed publication and presentations at relevant conferences.

## Supporting information

**S1 Appendix. Search strategy developed for MEDLINE (EBSCOhost) (20th May 2024).**
(DOCX)

**S2 Appendix. Data extraction instrument.**
(DOCX)

**S3 Appendix. PRISMA-P (Preferred Reporting Items for Systematic Review and Meta-Analysis Protocols) 2015 checklist.**
(DOCX)

## Acknowledgments

The authors acknowledge and thank librarian Katherine Miller and Aubrey Geyer at the University of British Columbia for their assistance.

## Author Contributions

**Conceptualization:** Lillian Hung.

**Writing – original draft:** Joey Wong, Haopu Ren, Yong Zhao.

**Writing – review & editing:** Lillian Hung, W. Ben Mortenson, Angelica Lim, Jennifer Boger, Christine Wallsworth.

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
