## [Decision Letter · Decision Letter 0]

7 Aug 2024

PONE-D-24-23780The use and impact of virtual reality programs supportedby aromatherapyfor older adults: A scoping review protocolPLOS ONE

Dear Dr. Hung,

Thank you for submitting your manuscript to PLOS ONE. After careful consideration, we feel that it has merit but does not fully meet PLOS ONE’s publication criteria as it currently stands. Therefore, we invite you to submit a revised version of the manuscript that addresses the points raised during the review process.

We look forward to receiving your revised manuscript.

Kind regards,

Rashid Menhas, PhD

Academic Editor

PLOS ONE

Journal Requirements:

One or more of the reviewers has recommended that you cite specific previously published works. Members of the editorial team have determined that some of the works referenced are not directly related to the submitted manuscript. As such, please note that it is not necessary or expected to cite the works requested by the reviewer.

Additional Editor Comments:

Dear Author (s)

Both reviewers comments are attached. You have to reply one by one comments of each reviewer in a response letter. Moreover, changes need to highlight in the revised manuscript.

Reviewers' comments:

Reviewer's Responses to Questions

**Comments to the Author**

1. Does the manuscript provide a valid rationale for the proposed study, with clearly identified and justified research questions?

Reviewer #1: Yes

Reviewer #2: Partly

2. Is the protocol technically sound and planned in a manner that will lead to a meaningful outcome and allow testing the stated hypotheses?

Reviewer #1: Yes

Reviewer #2: No

3. Is the methodology feasible and described in sufficient detail to allow the work to be replicable?

Reviewer #1: Yes

Reviewer #2: No

4. Have the authors described where all data underlying the findings will be made available when the study is complete?

Reviewer #1: Yes

Reviewer #2: Yes

5. Is the manuscript presented in an intelligible fashion and written in standard English?

Reviewer #1: Yes

Reviewer #2: No

6. Review Comments to the Author

You may also provide optional suggestions and comments to authors that they might find helpful in planning their study.

Reviewer #1: I have had the privilege of examining your work entitled " The use and impact of virtual reality programs supported by aromatherapy for older adults: A scoping review protocol" I acknowledge the research offered in the paper and compliment the thoroughness of your investigation into the subject topic. In general, I consider the work to be of high quality and directly applicable to the subject. I propose making some changes and providing recommendations for improvement.

1. Clarity in Terminology:

Ensure precision in the vocabulary used throughout the document. Although the majority of the information is well described, several concepts might be better understood by readers who are not very acquainted with the topic if they were given more explanation or context. Furthermore, substantiate their explanation using the most recent research findings. Please refer to the recommended readings for a more comprehensive understanding.

Nevertheless, the writers are recommended to tackle the following concerns before the ultimate submission and emphasize them in red for thorough verification.

The introductory portion is insufficiently lengthy and it is crucial to enhance its content. To enhance the quality of this research, it is crucial to include references to relevant works in both the introductory, literature review and discussion section and highlight all revisions for double checking.

Suggestions are:

1. https://doi.org/10.3389/fpsyg.2022.933974

2. https://doi.org/10.3389/fpsyg.2022.948061

3. https://doi.org/10.2147/PRBM.S441395

4. https://doi.org/10.3389/fpubh.2024.1228271

5. https://doi.org/10.2147/PRBM.S369020

6. https://doi.org/10.3389/fpubh.2023.1170645

7. https://doi.org/10.3389/fpubh.2024.1228271

The analysis should clearly articulate the influence of your findings with current research and provide a connection to the topic or question presented in the introduction. Verify assertions with substantiating facts and elucidate intricate arguments.

• Composing the conversation might be a demanding but fulfilling undertaking.

The capacity to analyze, establish connections, and integrate facts, while also producing a research output that has undergone peer review and dissemination, may be fulfilling. This is intended for the analysis and explanation of the main findings, as well as to emphasize the originality and importance of the research.

The discussion portion is very lengthy. It is necessary to differentiate the conclusion part and also write research implications.

Additionally, please outline the constraints of this research.

There are certain ambiguous and repetitive phrases.

Reviewer #2: The article has scientific potential, and includes a current theme.

1. The justification is weak, and does not adequately describe the articles already published, and the authors must show the relevance of the present study compared to the studies already published.

2. I recommend re-reading the references that guide scoping reviews, and also this recently published protocol. The search strategy must be expanded and systematized, as scoping reviews seek to carry out a broad mapping of the literature.

MDJ, P, Godfrey, C, McInerney, P, Baldini Soares, C, Khalil, H, and Parker, D. Chapter 11: scoping reviews In: E Aromataris and Z Munn, editors. JBI manual for evidence synthesis. Adelaide: JBI (2020); Pollock, D, Peters, MDJ, Khalil, H, McInerney, P, Alexander, L, Tricco, AC, et al. Recommendations for the extraction, analysis, and presentation of results in scoping reviews. JBI Evid Synth. (2023) 21:520–32. doi: 10.11124/JBIES-22-00123; Xavier PB, Silva ÍdS, Dantas THdM, Lopes RH, de Araújo AJ, de Figueirêdo RC and Uchôa SAdC (2024) Patient satisfaction and digital health in primary health care: a scoping review protocol. Front. Public Health. 12:1357688. doi: 10.3389/fpubh.2024.1357688.

3. Why will you include studies with people under the age of 65?

4. In the title: "supportedby" to "supported by"

7. PLOS authors have the option to publish the peer review history of their article (what does this mean?). If published, this will include your full peer review and any attached files.

Reviewer #1: No

Reviewer #2: **Yes: **Isis de Siqueira Silva

---

## [Author Response · Author response to Decision Letter 0]

16 Aug 2024

Reviewers' comments:

Reviewer's Responses to Questions

Comments to the Author

1. Does the manuscript provide a valid rationale for the proposed study, with clearly identified and justified research questions?

Reviewer #1: Yes

Reviewer #2: Partly

• The introduction section provides the rationale for this scoping review, highlighting the potentials of integrating olfactory stimuli in VR programs. The main rationale of the study is to address the gap of a lack of review on VR supported by aromatherapy in older adults. To date, there has been no scoping review reported on VR programs supported by aromatherapy. We will summarize the evidence regarding VR programs with aromatherapy for older adults, including the outcome measures and outcomes 

2. Is the protocol technically sound and planned in a manner that will lead to a meaningful outcome and allow testing the stated hypotheses?

Reviewer #1: Yes

Reviewer #2: No

• The scoping review does not involve experimental procedure, analysis pipeline, proposed hypotheses and statistical power analysis. This scoping review follows JBI methodology, and no analyses are exploratory.

3. Is the methodology feasible and described in sufficient detail to allow the work to be replicable?

Reviewer #1: Yes

Reviewer #2: No

• This manuscript is a scoping review protocol, which does not need controls, sample size calculations, and replication.

4. Have the authors described where all data underlying the findings will be made available when the study is complete?

Reviewer #1: Yes

Reviewer #2: Yes

5. Is the manuscript presented in an intelligible fashion and written in standard English?

Reviewer #1: Yes

Reviewer #2: No

• This manuscript was prepared by several authors whose first language is English and has been professionally checked to ensure no typographical or grammatical errors .

6. Review Comments to the Author

You may also provide optional suggestions and comments to authors that they might find helpful in planning their study.

Reviewer #1: I have had the privilege of examining your work entitled " The use and impact of virtual reality programs supported by aromatherapy for older adults: A scoping review protocol" I acknowledge the research offered in the paper and compliment the thoroughness of your investigation into the subject topic. In general, I consider the work to be of high quality and directly applicable to the subject. I propose making some changes and providing recommendations for improvement.

1. Clarity in Terminology:

Ensure precision in the vocabulary used throughout the document. Although the majority of the information is well described, several concepts might be better understood by readers who are not very acquainted with the topic if they were given more explanation or context. Furthermore, substantiate their explanation using the most recent research findings. Please refer to the recommended readings for a more comprehensive understanding.

Nevertheless, the writers are recommended to tackle the following concerns before the ultimate submission and emphasize them in red for thorough verification.

The introductory portion is insufficiently lengthy and it is crucial to enhance its content. To enhance the quality of this research, it is crucial to include references to relevant works in both the introductory, literature review and discussion section and highlight all revisions for double checking.

Suggestions are:

1. https://doi.org/10.3389/fpsyg.2022.933974

2. https://doi.org/10.3389/fpsyg.2022.948061

3. https://doi.org/10.2147/PRBM.S441395

4. https://doi.org/10.3389/fpubh.2024.1228271

5. https://doi.org/10.2147/PRBM.S369020

6. https://doi.org/10.3389/fpubh.2023.1170645

7. https://doi.org/10.3389/fpubh.2024.1228271

The analysis should clearly articulate the influence of your findings with current research and provide a connection to the topic or question presented in the introduction. Verify assertions with substantiating facts and elucidate intricate arguments.

• Composing the conversation might be a demanding but fulfilling undertaking.

The capacity to analyze, establish connections, and integrate facts, while also producing a research output that has undergone peer review and dissemination, may be fulfilling. This is intended for the analysis and explanation of the main findings, as well as to emphasize the originality and importance of the research.

The discussion portion is very lengthy. It is necessary to differentiate the conclusion part and also write research implications.

Additionally, please outline the constraints of this research.

There are certain ambiguous and repetitive phrases.

• The manuscript we submitted is a scoping review protocol, not a research study. We do not have findings, discussion, and conclusion so we are not sure about the comments on those parts. 

• For terminology, we have now expanded the description on several systematic reviews to provide more content for background explanation.

• The literature suggested by reviewer related to student learning, social prescription, health impact of COVID-19, which do not relate to our topic “combination of VR and aromatherapy”. Below is the literature suggested by the reviewer:

• Noor, U., Younas, M., Saleh Aldayel, H., Menhas, R., & Qingyu, X. (2022). Learning behavior, digital platforms for learning and its impact on university student's motivations and knowledge development. Frontiers in Psychology, 13, 933974-933974. https://doi.org/10.3389/fpsyg.2022.933974

• Younas, M., Noor, U., Zhou, X., Menhas, R., & Qingyu, X. (2022). COVID-19, students satisfaction about e-learning and academic achievement: Mediating analysis of online influencing factors. Frontiers in Psychology, 13, 948061-948061. https://doi.org/10.3389/fpsyg.2022.948061

• Younas, M., Dong, Y., Menhas, R., Li, X., Wang, Y., & Noor, U. (2023). Alleviating the effects of the COVID-19 pandemic on the physical, psychological health, and wellbeing of students: Coping behavior as a mediator. Psychology Research and Behavior Management, 16, 5255-5270. https://doi.org/10.2147/PRBM.S441395

• Menhas, R., Yang, L., Saqib, Z. A., Younas, M., & Saeed, M. M. (2024). Does nature-based social prescription improve mental health outcomes? A systematic review and meta-analysis. Frontiers in Public Health, 12, 1228271-1228271. https://doi.org/10.3389/fpubh.2024.1228271

• Peng, X., Menhas, R., Dai, J., & Younas, M. (2022). The COVID-19 pandemic and overall wellbeing: Mediating role of virtual reality fitness for physical-psychological health and physical activity. Psychology Research and Behavior Management, 15, 1741-1756. https://doi.org/10.2147/PRBM.S369020

• Menhas, R., Qin, L., Saqib, Z. A., & Younas, M. (2023). The association between COVID-19 preventive strategies, virtual reality exercise, use of fitness apps, physical, and psychological health: Testing a structural equation moderation model. Frontiers in Public Health, 11, 1170645-1170645. https://doi.org/10.3389/fpubh.2023.1170645

• Menhas, R., Yang, L., Saqib, Z. A., Younas, M., & Saeed, M. M. (2024). Does nature-based social prescription improve mental health outcomes? A systematic review and meta-analysis. Frontiers in Public Health, 12, 1228271-1228271. https://doi.org/10.3389/fpubh.2024.1228271

Reviewer #2: The article has scientific potential, and includes a current theme.

1. The justification is weak, and does not adequately describe the articles already published, and the authors must show the relevance of the present study compared to the studies already published.

• Thanks and we appreciate this comment. We have now provided more description of several articles recently published (1,3,6,13,18,25) and highlighted the potentials of integrating olfactory stimuli in VR programs. The main rationale of the study is to address the gap of a lack of review on VR supported by aromatherapy in older adults. To date, there has been no scoping review reported on VR programs supported by aromatherapy. Current VR reviews do not focus on the combination of VR and aromatherapy, just focus on the impacts of VR [1,3,6,8] or the impacts of aromatherapy [13,18,25] separately, which emphasizes the research gap needs to be covered by this scoping review and further research. We will summarize the evidence regarding VR programs with aromatherapy for older adults, including the outcome measures and outcomes.

2. I recommend re-reading the references that guide scoping reviews, and also this recently published protocol. The search strategy must be expanded and systematized, as scoping reviews seek to carry out a broad mapping of the literature.

MDJ, P, Godfrey, C, McInerney, P, Baldini Soares, C, Khalil, H, and Parker, D. Chapter 11: scoping reviews In: E Aromataris and Z Munn, editors. JBI manual for evidence synthesis. Adelaide: JBI (2020); Pollock, D, Peters, MDJ, Khalil, H, McInerney, P, Alexander, L, Tricco, AC, et al. Recommendations for the extraction, analysis, and presentation of results in scoping reviews. JBI Evid Synth. (2023) 21:520–32. doi: 10.11124/JBIES-22-00123; 

Xavier PB, Silva ÍdS, Dantas THdM, Lopes RH, de Araújo AJ, de Figueirêdo RC and Uchôa SAdC (2024) Patient satisfaction and digital health in primary health care: a scoping review protocol. Front. Public Health. 12:1357688. doi: 10.3389/fpubh.2024.1357688.

• Thanks for providing the two resources. The first reference provides valuable guidance on synthesis and the presentation of results, which is instrumental for our process. We will be following the process in reporting the results in the next paper – the scoping review results. 

• The second reference is a published scoping review protocol with a search strategy similar to ours. We appreciate the example. MedRxiv, an online disciplinary repository that publishes preprints and is indexed by Google Scholar and Web of Science, is already within the search scope of our protocol. Regarding gray literature, we have included theses and dissertations via the ProQuest database, conference proceedings through the Web of Science database, and additional gray literature via Google Scholar. Furthermore, we have expanded our search to include the ACM Digital Library, IEEE Xplore Digital Library, and Compendex database. These databases focus on a broad range of topics in computing, engineering, and technology fields, ensuring that we capture potential literature on Virtual Reality (VR). In addition, we are collaborating with two medical librarians to refine the search strategy to ensure we capture all relevant and key articles.

3. Why will you include studies with people under the age of 65?

• This scoping review will only include studies which recruited older adults whose age is older than 65.

4. In the title: "supportedby" to "supported by"

• We checked and avoided words accidentally concatenated.

Journal Requirements:

• The manuscript has now been checked to meet style requirements and author affilications requirement.

• Data availability statement has been added at the end of the manuscript before references section.

One or more of the reviewers has recommended that you cite specific previously published works. Members of the editorial team have determined that some of the works referenced are not directly related to the submitted manuscript. As such, please note that it is not necessary or expected to cite the works requested by the reviewer.

• The reference list has been reviewed now to ensure its accuracy.

---

## [Decision Letter · Decision Letter 1]

27 Nov 2024

PONE-D-24-23780R1The use and impact of virtual reality programs supported by aromatherapy for older adults: A scoping review protocolPLOS ONE

Dear Dr. Hung,

Thank you for submitting your manuscript to PLOS ONE. After careful consideration, we feel that it has merit but does not fully meet PLOS ONE’s publication criteria as it currently stands. Therefore, we invite you to submit a revised version of the manuscript that addresses the points raised during the review process.

We look forward to receiving your revised manuscript.

Kind regards,

Muhammad Shahid Anwar

Academic Editor

PLOS ONE

Comments from PLOS Editorial Office: We note that Reviewer 1 has recommended that you cite specific previously published works in an earlier round of revision. Members of the editorial team have determined that the works referenced are not directly related to the submitted manuscript. As such, please note that it is not necessary or expected to cite the works requested by the reviewer. 

Additional Editor Comments:

Comments

1. In the abstract, try to avoid repeating words in a single sentence (outcomes) for example, ….”” “Therefore, this scoping 26 review will identify existing types of virtual reality programs supported by various forms of 27 aromatherapies and their outcomes and outcome measures on the well-being of older adults. “”

2. Also, it is mentioned that the review will identify outcomes and outcome measures but does not specify examples. Including 1-2 key examples of anticipated outcomes (e.g., stress reduction, cognitive benefits, etc)

3. The introduction covers a wide range of topics (VR, aromatherapy, and their integration) but it lacks a smoother transition between these topics. For example, the shift between the sections on VR and aromatherapy feels abrupt. I suggest adding a linking sentence to explain how these topics are interconnected.

4. There should also be a statement of “whether VR has been used to improve mobility or where aromatherapy reduced anxiety in older adults.

5. The contribution and novelty of the article is unclear. The authors should clearly state the contribtion and novelty of the article.

6. The contribution part should include “why understanding the combined impact of VR and aromatherapy is essential?”

7. Also the motivation of the article is not satisfying. What are the exact mootivation of this article.

8. The figures quality should be high some of them are blur.

Reviewers' comments:

Reviewer's Responses to Questions

**Comments to the Author**

1. Does the manuscript provide a valid rationale for the proposed study, with clearly identified and justified research questions?

Reviewer #1: No

Reviewer #2: Yes

2. Is the protocol technically sound and planned in a manner that will lead to a meaningful outcome and allow testing the stated hypotheses?

Reviewer #1: No

Reviewer #2: Yes

3. Is the methodology feasible and described in sufficient detail to allow the work to be replicable?

Reviewer #1: No

Reviewer #2: Yes

4. Have the authors described where all data underlying the findings will be made available when the study is complete?

Reviewer #1: No

Reviewer #2: Yes

5. Is the manuscript presented in an intelligible fashion and written in standard English?

Reviewer #1: No

Reviewer #2: Yes

6. Review Comments to the Author

You may also provide optional suggestions and comments to authors that they might find helpful in planning their study.

Reviewer #1: The authors are unwilling or unable to address my concerns sufficiently to make this manuscript suitable for publication.

Reviewer #2: The authors submitted the manuscript with the corrections. The authors have worked hard to improve the methodology of the study, allowing it to be clearly replicated,

7. PLOS authors have the option to publish the peer review history of their article (what does this mean?). If published, this will include your full peer review and any attached files.

Reviewer #1: No

Reviewer #2: No

---

## [Author Response · Author response to Decision Letter 1]

28 Nov 2024

Additional Editor Comments:

Comments

1. In the abstract, try to avoid repeating words in a single sentence (outcomes) for example, ….”” “Therefore, this scoping 26 review will identify existing types of virtual reality programs supported by various forms of 27 aromatherapies and their outcomes and outcome measures on the well-being of older adults. “”

Response: Thanks for your comments. We have revised “outcomes” into “results.”

2. Also, it is mentioned that the review will identify outcomes and outcome measures but does not specify examples. Including 1-2 key examples of anticipated outcomes (e.g., stress reduction, cognitive benefits, etc)

Response: Thank you for your comments. We had already identified a relevant piece of literature before conducting this scoping review, which provided examples of anticipated outcomes such as happiness, sleep quality, perceived stress, and life satisfaction (Page 9, Lines 118-120, Reference 24). In response to your suggestion, we have also included a description of these outcomes in the data synthesis section (Page 15, Lines 241-242)." 

3. The introduction covers a wide range of topics (VR, aromatherapy, and their integration) but it lacks a smoother transition between these topics. For example, the shift between the sections on VR and aromatherapy feels abrupt. I suggest adding a linking sentence to explain how these topics are interconnected.

Response: Thanks for your comments. We have added one linking sentence between these two topics. (Page 5, Line 63–65)

4. There should also be a statement of “whether VR has been used to improve mobility or where aromatherapy reduced anxiety in older adults.

Response: Thanks for your useful comments. We mentioned VR can improve physical health like mobility (Page 5, Line 50-51, references 4-6) and aromatherapy can reduce anxiety (Page 6, Line 71-72, reference 14 &15). 

5. The contribution and novelty of the article is unclear. The authors should clearly state the contribtion and novelty of the article.

Response: Thanks for your comment. We have revised the introduction section to address the novelty and contribution of this scoping review. (Page 9, Line 123-124; 127-128)

6. The contribution part should include “why understanding the combined impact of VR and aromatherapy is essential?”

Response: Thanks for your comments. We added this contribution to the introduction section (Page 9, Line 128-130).

7. Also the motivation of the article is not satisfying. What are the exact mootivation of this article.

Response: Thank you for your comments. After updating the novelty and contribution in the introduction section, it now effectively supports the motivation for conducting this scoping review. (Page 9, Line 121-130).

8. The figures quality should be high some of them are blur.

Response: We have no figure for this scoping review protocol.

Reviewers' comments:

Reviewer's Responses to Questions

Comments to the Author

1. Does the manuscript provide a valid rationale for the proposed study, with clearly identified and justified research questions?

Reviewer #1: No

Response: We apologize for the change in your comments from all 'YES' to all 'NO'; we understand that our efforts do not meet your requirements. Please refer to our response to Comment No. 6 for further clarification.

Reviewer #2: Yes

2. Is the protocol technically sound and planned in a manner that will lead to a meaningful outcome and allow testing the stated hypotheses?

Reviewer #1: No

Reviewer #2: Yes

3. Is the methodology feasible and described in sufficient detail to allow the work to be replicable?

Reviewer #1: No

Reviewer #2: Yes

4. Have the authors described where all data underlying the findings will be made available when the study is complete?

Reviewer #1: No

Reviewer #2: Yes

5. Is the manuscript presented in an intelligible fashion and written in standard English?

Reviewer #1: No

Reviewer #2: Yes

6. Review Comments to the Author

You may also provide optional suggestions and comments to authors that they might find helpful in planning their study.

Reviewer #1: The authors are unwilling or unable to address my concerns sufficiently to make this manuscript suitable for publication.

Response: We apologize that we cannot add findings, discussion, and conclusion in a protocol. We apologize that we cannot add nonrelevant references to this protocol. We do try our best to expand the description on several systematic reviews for sufficient background explanation. We are looking forward to your more constructive suggestions.

Reviewer #2: The authors submitted the manuscript with the corrections. The authors have worked hard to improve the methodology of the study, allowing it to be clearly replicated,

7. PLOS authors have the option to publish the peer review history of their article (what does this mean?). If published, this will include your full peer review and any attached files.

Do you want your identity to be public for this peer review? For information about this choice, including consent withdrawal, please see our Privacy Policy.

Reviewer #1: No

Reviewer #2: No

---

## [Editor Report · Decision Letter 2]

4 Dec 2024

PONE-D-24-23780R2The use and impact of virtual reality programs supported by aromatherapy for older adults: A scoping review protocolPLOS ONE

Dear Dr. Hung,

Thank you for submitting your manuscript to PLOS ONE. After careful consideration, we feel that it has merit but does not fully meet PLOS ONE’s publication criteria as it currently stands. Therefore, we invite you to submit a revised version of the manuscript that addresses the points raised during the review process.

We look forward to receiving your revised manuscript.

Kind regards,

Muhammad Shahid Anwar

Academic Editor

PLOS ONE

Journal Requirements:

Additional Editor Comments:

1. The author should clearly mention the contribution in bullets at the end of the Introduction Section.

2. There should also be objectives of the study for easy understanding of readers.

3. Thereshould also be indiation of each review question that mention the section where each question is addressed.

---

## [Author Response · Author response to Decision Letter 2]

4 Dec 2024

Additional Editor Comments:

1. The author should clearly mention the contribution in bullets at the end of the Introduction Section.

Response: Thanks for your comments. The contribution in bullets is added at the end of the introduction section behind the “review question.” (Page 8, Line 139-150)

2. There should also be objectives of the study for easy understanding of readers.

Response: Thanks for your comments. The objectives are revised to be mentioned in two bullets for easy understanding of readers. (Page 7, Line 123, 125)

3. There should also be indication of each review question that mention the section where each question is addressed.

Response: Thanks for your suggestion. We added an explanation of the indication of each review question in the method section “Data synthesis” (Pages 13-14, Line 259 – 266). The revision ensures that each review question is clearly addressed, making the findings more oriented.

---

## [Editor Report · Decision Letter 3]

19 Dec 2024

The use and impact of virtual reality programs supported by aromatherapy for older adults: A scoping review protocol

PONE-D-24-23780R3

Dear Dr. Hung,

We’re pleased to inform you that your manuscript has been judged scientifically suitable for publication and will be formally accepted for publication once it meets all outstanding technical requirements.

Kind regards,

Muhammad Shahid Anwar

Academic Editor

PLOS ONE

Additional Editor Comments (optional):

The authors have addressed all the comments properly. Therefore the article is ready to accept. However, the authors are advised to review the article and correct the grammer if needed.
---

## [Editor Report · Acceptance letter]

29 Dec 2024

PONE-D-24-23780R3 

PLOS ONE

Dear Dr. Hung, 

I'm pleased to inform you that your manuscript has been deemed suitable for publication in PLOS ONE. Congratulations! Your manuscript is now being handed over to our production team.

Kind regards, 

on behalf of

Professor Muhammad Shahid Anwar 

Academic Editor

PLOS ONE